# Synthesis of Gd₂O₃ Nanoparticles and Their Photocatalytic Activity for Degradation of Azo Dyes

Sugyeong Jeon [1], Jeong-Won Ko [2] and Weon-Bae Ko [1,2,3,*]

[1] Department of Convergence Science, Graduate School of Sahmyook University, Seoul 01795, Korea; kowb1@naver.com

[2] Department of Animal Life Resources, Chemistry Major, Sahmyook University, 815 Hwarang-ro, Nowon-gu, Seoul 01795, Korea; jwko7121@naver.com

[3] Department of Chemistry, Sahmyook University, Seoul 01795, Korea

[*] Correspondence: kowb@syu.ac.kr; Tel.: +82-02-3399-1700

**Abstract:** Gadolinium oxide (Gd₂O₃) nanoparticles were prepared via the reaction of gadolinium nitrate hexahydrate (Gd (NO₃)₃·6H₂O) and ethylamine (C₂H₅NH₂), and their surface morphology, particle size, and properties were examined by using scanning electron microscopy, X-ray diffraction (XRD), Raman spectroscopy, Fourier transform infrared (FT-IR) spectroscopy, and ultraviolet visible (UV-vis) spectroscopy. The Gd₂O₃ nanoparticles were used as the photocatalyst for the degradation of various azo dyes, such as methyl orange (MO), acid orange 7 (AO7), and acid yellow 23 (AY23) under irradiation with UV light. The effect of the experimental parameters (initial concentration of azo dyes, dosage of catalyst, and wavelength of UV light) on the photocatalytic properties of the Gd₂O₃ nanoparticles were investigated. At a constant H₂O₂ concentration, the photocatalytic degradation efficiency of the Gd₂O₃ nanoparticles for various azo dyes was in the order: methyl orange > acid orange 7 > acid yellow 23. The kinetics study showed that the photocatalytic degradation of azo dyes was followed by a pseudo first-order reaction rate law.

**Keywords:** gadolinium oxide nanoparticles; photocatalytic activity; azo dyes; kinetics study





## 1. Introduction

The use of a metal oxide semiconductor photocatalyst is a common technique for the degradation of organic pollutants [1,2]. The textile industry generates a lot of wastewater, which contains a lot of dyes. Many contaminants in the ecosystem have been problematic due to their stable chemical structure, posing various teratogenic, carcinogenic, and mutagenic risks [3]. As a result, wastewater purification has always received a lot of interest in the scientific field. The combined advanced oxidation process (AOP), which includes heterogeneous photocatalysis and hydrogen peroxide, provides the synergistic effect for degradation of organic dyes in wastewater [3–7]. The dye degradation mechanism of the AOP involves the generation of hydroxyl radicals (·OH) and superoxide anion radicals (·O₂⁻) [8–10]. Their radicals are unstable and can attack organic pollutants to make harmless products.

Lanthanide compounds have been used in photocatalysis, owing to their unique *f*-electronic configuration and bandgap energy. Rare earth elements have half-filled 4*f*-shells with unpaired electrons and often exhibit an empty 5*d* shell. They are used in a wide range of applications, such as fluorescent materials, high-resolution X-ray medical imaging, ultraviolet (UV) detectors, catalysts, and dopants [11–16]. Owing to its high chemical stability, UV absorption, and active photon-to-electron conversion, gadolinium oxide (Gd₂O₃) is widely used as an *n*-type semiconductor photocatalyst [12,17,18].

Recently, several papers have been reported on photocatalytic degradation by using lanthanide oxide nanoparticles [11,12,15,16]. However, there are still insufficient studies available on the degradation of lanthanide oxide nanoparticles of organic pollutants.

$Gd_2O_3$ nanoparticles can be used for the photocatalytic degradation of organic pollutants and wastewater treatment, and, due to their high photocatalytic activity, they are also used as an additional oxidizing agent along with other agents, such as hydrogen peroxide [19–21].

In $Gd_2O_3$, the electrons in the valence band move to the conduction band under UV irradiation [22]. Gadolinium oxide nanoparticles effectively produce free radicals and their recombination with holes is avoided because of the presence of half-filled 4*f* orbitals and an empty 5*d* shell, which often serve as trapping centers to prevent the combination of carriers [23]. The ·OH radicals generated in water under UV irradiation are responsible for the degradation of azo dyes [19]. The unique 4*f* orbitals/electrons of rare earth compounds have magnetic, electrical, optical, phosphorus, and catalytic properties [17,24].

AOPs are based on the combination of strong oxidants with a catalyst and radiation; the addition of hydrogen peroxide increases the rate [25]. The photocatalytic degradation efficiency of organic dyes may be enhanced in combination with $H_2O_2$ and $Gd_2O_3$ nanoparticles under UV irradiation. ln this study, we carried out photocatalytic degradation of various organic dyes, such as methyl orange (MO), acid orange 7 (AO7), and acid yellow 23 (AY23) under different conditions using the $Gd_2O_3$ nanoparticles.

## 2. Results

### 2.1. Characterization of $Gd_2O_3$ Nanoparticles

For $Gd_2O_3$, the XRD peaks observed at $2\theta = 28.64°$ correspond to the (222) plane of the cubic phase (JCPDS 3-065-3181) [24], and those at $2\theta = 20.14°, 33.16°, 47.64°$, and $56.53°$ correspond to the (211), (400), (440), and (622) planes, respectively, of the hexagonal phase (Figure 1). The size of the synthesized $Gd_2O_3$ nanoparticles was calculated using the Scherrer formula:

$$D = \frac{K\lambda}{\beta cos\theta} \tag{1}$$

where $D$ is the size of the crystallites, $K$ is the Scherrer constant, $\lambda$ is the X-ray wavelength, $\beta$ is the full width at half maximum (in radians) of the XRD peak, and $\theta$ is the Bragg angle. The average crystallite size of the $Gd_2O_3$ nanoparticles was calculated to be 13.7 nm [24].

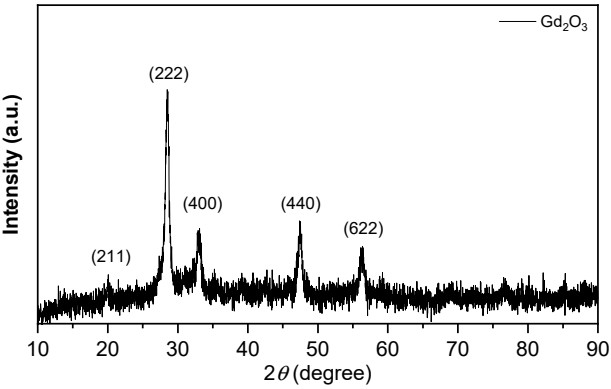

**Figure 1.** XRD pattern of the synthesized $Gd_2O_3$ nanoparticles.

XPS analysis determined the surface elemental composition and their ionic states. Figure 2a depicts the appearance of Gd 3d, Gd 4d, and O 1s elements of the $Gd_2O_3$ nanoparticles in the XPS survey spectrum. The binding energy at 1187.9 and 1219.3 eV corresponds to the Gd $3d_{5/2}$ and Gd $3d_{3/2}$, respectively, in Figure 2b [12,18]. Figure 2c demonstrates that the binding energy at 142.2 eV and 147.8 eV were ascribed to the Gd $4d_{5/2}$ and Gd $4d_{3/2}$ spin-orbits, respectively. As for Figure 2d, the O 1s peak in the $Gd_2O_3$ nanoparticles displayed the binding energy at 531.7 eV [19].

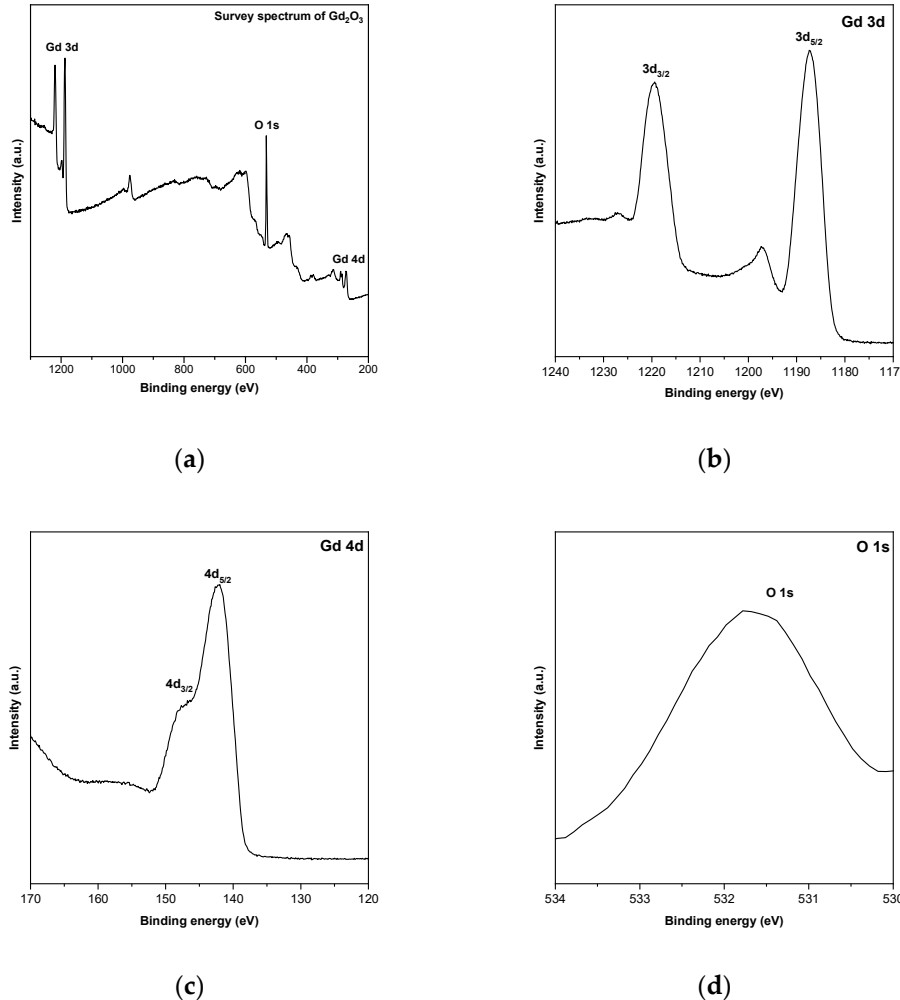

**Figure 2.** XPS spectra of the $Gd_2O_3$ nanoparticles (**a**) full scan survey spectrum, (**b**) Gd 3d, (**c**) Gd 4d and (**d**) O 1s.

The SEM image of $Gd_2O_3$ in Figure 3a have a pine needle-like shape. The TEM image of $Gd_2O_3$ in Figure 3b shows a rod shape with the length of $100-900$ nm. Additionally, in Figure 3c,d, the elemental compositions of the synthesized compounds were determined by EDX mapping/elemental analysis. According to the EDX spectrum, the $Gd_2O_3$ nanoparticles contained a gadolinium atom (Gd) of 34.31%, and an oxygen atom (O) of 65.69%, as shown in Figure 3e [15,19].

The vibrational modes of $Gd_2O_3$ nanoparticles were observed by Raman spectroscopy in Figure 4a. In the Raman spectrum, three peaks were observed: two at 313 and 443 cm$^{-1}$, corresponding to the $F_g + E_g$ modes and the $F_g$ mode, respectively, and most importantly, the prominent peak at 358 cm$^{-1}$ is assigned to the $F_g + A_g$ modes, which mainly corresponds to the cubic phase of the $Gd_2O_3$ nanoparticles [26].

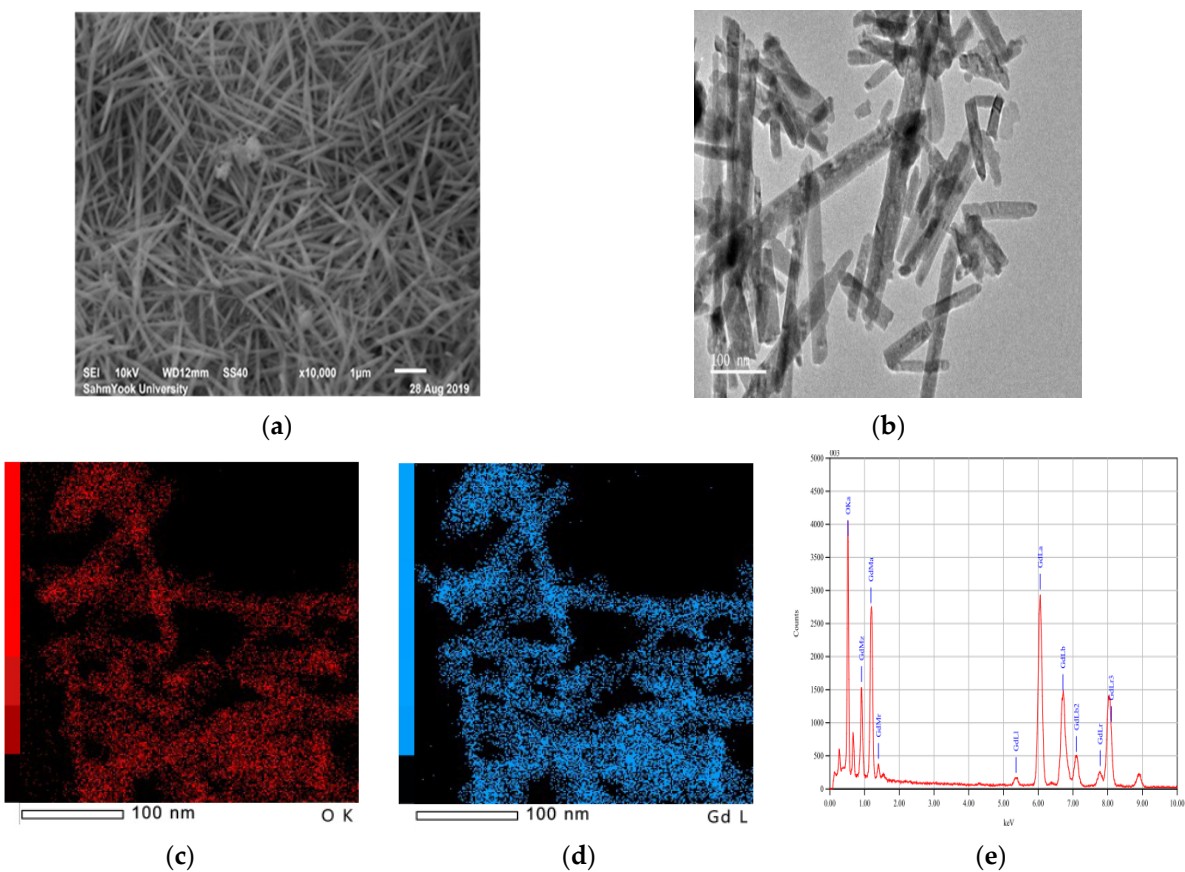

**Figure 3.** Morphological analysis (**a**) SEM images and (**b**) TEM images of the $Gd_2O_3$ nanoparticles. Elemental analysis (**c**) and (**d**) EDX mapping composition of the $Gd_2O_3$ nanoparticles, (**e**) EDX spectrum of the $Gd_2O_3$ nanoparticles, they are kind of elements such as Gd, O.

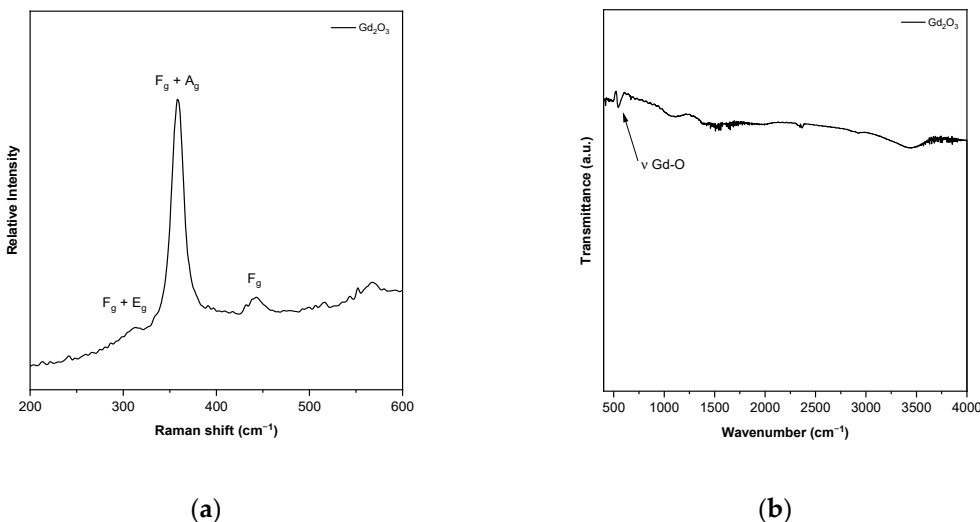

**Figure 4.** (**a**) Raman spectrum and (**b**) FT-IR spectrum of the $Gd_2O_3$ nanoparticles.

The $Gd_2O_3$ nanoparticles were investigated by FT-IR spectroscopy. The $Gd_2O_3$ nanoparticles show the characteristic peak at 547 $cm^{-1}$ due to the banding vibration of Gd-O in the FT-IR spectroscopy in Figure 4b [23,24].

The absorbance peak of the $Gd_2O_3$ nanoparticles was observed at 228 nm in UV-vis spectroscopy [27]. The optical bandgap energy ($E_g$) was estimated by the method proposed by Tauc [17]. The optical band gap of $Gd_2O_3$ is determined by the following equation:

$$\alpha h\nu = A\left[h\nu - E_g\right]^k \tag{2}$$

where $\alpha$ is absorption coefficient, $h$ is Plank constant, $\nu$ is light frequency, $A$ is a characteristic constant of the semiconductor, $E_g$ is the apparent optical band gap of the material, and $k$ is a constant associated with electronic transition types (for direct allowed transition: $k = 1/2$, for direct forbidden transitions: $k = 2/3$, for indirect allowed transition: $k = 2$, and for indirect forbidden transitions: $k = 3$). According to the literature [23,27], gadolinium oxide nanoparticles were characterized by a direct allowed electronic transition, so $k = 1/2$ was used. The value of $E_g$ was evaluated by extrapolating the linear portion of the curve in Figure 5b. This was found to be 5.04 eV for the $Gd_2O_3$ nanoparticles.

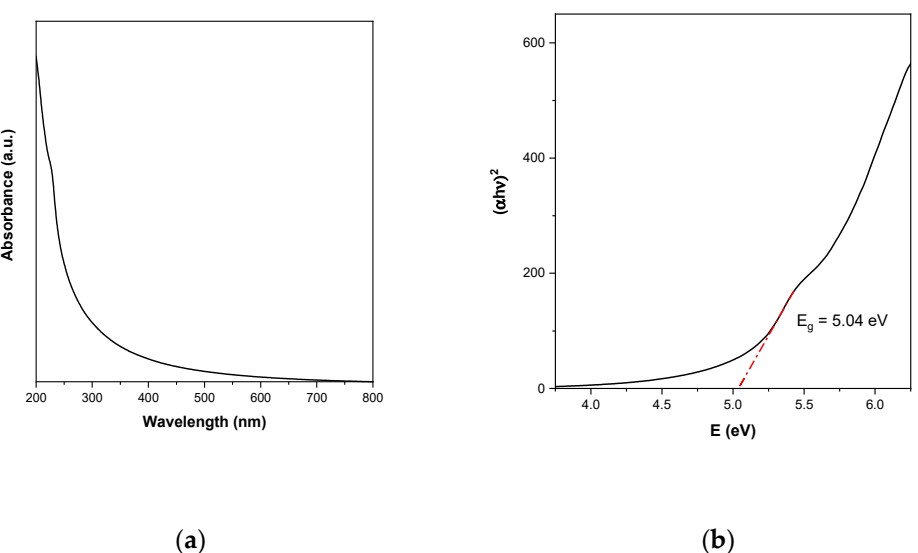

(**a**)  (**b**)

**Figure 5.** (**a**) UV-vis spectrum and (**b**) Tauc's plot for the bandgap energy of the $Gd_2O_3$ nanoparticles.

When aqueous solutions of azo dyes were irradiated by UV light without adding a catalyst, the azo dye degradation process was very slow. The addition of the $Gd_2O_3$ nanoparticles, i.e., the catalyst, accelerated the azo dye degradation process.

The aqueous solution of MO containing the $H_2O_2$ and $Gd_2O_3$ nanoparticles showed an absorbance peak at 480 nm (Figure 6a). During the photocatalytic degradation reaction, no other absorbance peak was observed. The absorbance peak in the visible region faded away after UV irradiation at 254 nm for 100 min. The absorption peak appeared at 480 nm, and the intensity of this peak decreased with UV light irradiation. With time, this peak shifted to a lower absorbance in the visible region. The MO showed a degradation of 79.68% (Figure S1c) and 26.62% (Figure S1a) in the presence and absence of the catalyst, respectively.

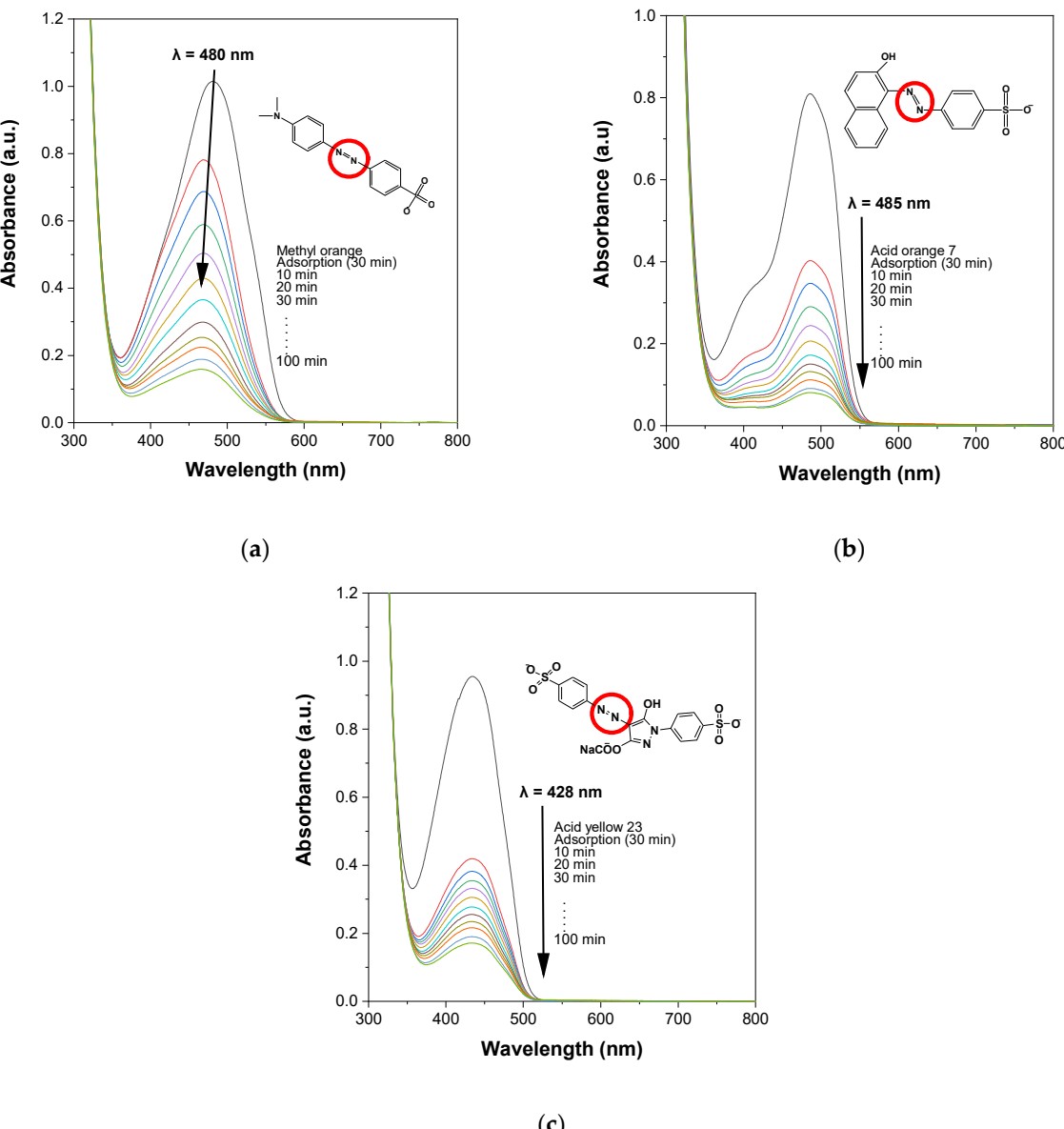

**Figure 6.** UV-vis absorption spectra of photocatalytic degradation of the azo dyes (**a**) MO, (**b**) AO7, and (**c**) AY23 by the $Gd_2O_3/H_2O_2/UV$ method for 100 min. (Experimental conditions: initial dye concentration: 0.042 mM; photocatalyst concentration: 1.0 g/L; $H_2O_2$ concentration: 4.8M; wavelength of UV irradiation: 254 nm).

The absorbance peak of AO7 appeared at 485 nm (Figure 6b). After 100 min of UV irradiation in the absence of the catalyst, only 6.11% (Figure S2a) of the AO7 dye was degraded. On the other hand, a 79.92% (Figure S2c) degradation was achieved in the presence of the $Gd_2O_3$ catalyst.

AY23 showed an absorption peak at 428 nm (Figure 6c). The AY23 was the active site for an oxidative attack. The AY23 showed a degradation of 59.14% (Figure S3c) and 3.81% (Figure S3a) in the presence and absence of $Gd_2O_3$ catalyst, respectively.

The adsorption of $H_2O_2$ onto $Gd_2O_3$ particles can modify their surface [28–30]. The dye adsorption capacity on metal oxides is an important parameter for surface reactions; it depends on the surface area and other surface-related parameters, such as the surface density, pore size distribution, and morphology of the particle surface. If the adsorption of dye molecules increases, the rate of photocatalytic degradation increases. The photocatalytic degradation efficiency of semiconductor oxides under UV irradiation is related to the electronic band structure of the semiconductor.

The histograms of photocatalytic degradation efficiency (MO, AO7, AY23) under various conditions (UV/$H_2O_2$, $H_2O_2$/$Gd_2O_3$, UV/$H_2O_2$/$Gd_2O_3$, and UV/$Gd_2O_3$ for 100 min) are shown in Figure 7. It can be observed from the spectra that the $Gd_2O_3$ nanoparticles accelerated the degradation of the azo dyes. The photocatalytic degradation efficiency was the highest when both $Gd_2O_3$ and $H_2O_2$ were used under UV irradiation conditions.

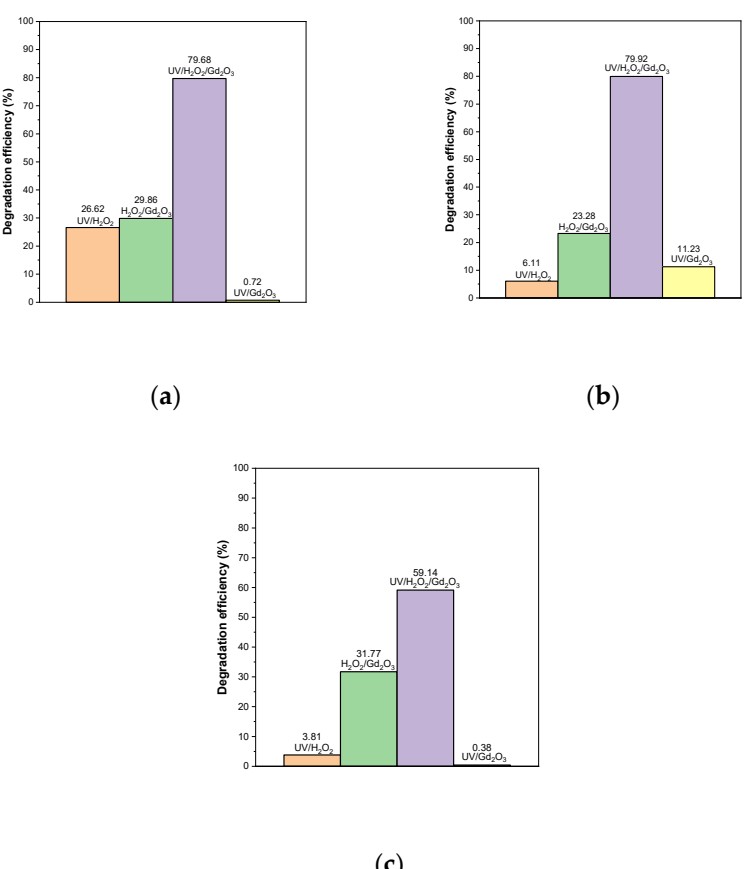

**Figure 7.** Histograms of photocatalytic degradation efficiency of (**a**) MO, (**b**) AO7, and (**c**) AY23 under different conditions after 100 min. (Experimental conditions: initial dye concentration: 0.042 mM; photocatalyst concentration: 1.0 g/L; $H_2O_2$ concentration: 4.8M; wavelength of UV irradiation: 254 nm).

The photocatalytic degradation efficiency of azo dyes using $Gd_2O_3$ is calculated by the following equation:

$$\text{Photocatalytic degradation efficiency } (\%) = \frac{(C_0 - C_t)}{C_0} \times 100 \qquad (3)$$

where $C_0$ is the concentration of azo dye at the adsorption-desorption equilibrium in the dark ($t = 0$) and $C_t$ is the concentration of azo dye at reaction time $t$ (min).

## 2.2. Mechanism of Photocatalytic Degradation of Azo Dyes by $Gd_2O_3$ Nanoparticles

Upon UV irradiation, the electrons in the valence band of the catalyst moved into the conduction band. This resulted in the continuous generation of holes ($h^+$) in the valence band and electrons ($e^-$) in the conduction band. The generation of electron-hole pairs contributed to the activity of the photocatalyst. The holes with a high oxidative ability can oxidize $OH^-$ to $\cdot OH$ radicals. The holes in the valence band and electrons in the conduction band were generated and combined with a hydroxide ion and oxygen to generate $\cdot OH$ and $\cdot O^{2-}$ radicals. The photocatalytic degradation of the organic dyes occurred on the

surface of the catalyst. The ·OH and ·$O_2^-$ radicals act as oxidizing agents for azo dye molecules [19].

The azo dye degradation mechanism under the UV/$H_2O_2$/$Gd_2O_3$ condition is as follows (Figure 8) [12,15,19]:

$$Gd_2O_3 + h\nu \ \rightarrow \ e_{cb}^- + h_{vb}^+ \tag{4}$$

$$H_2O_2 + h\nu \ \rightarrow \ 2 \cdot OH \tag{5}$$

$$O_2 + Gd_2O_3(e_{cb}^-) \rightarrow \cdot O_2^- \tag{6}$$

$$H_2O_2 + Gd_2O_3(e_{cb}^-) \rightarrow \cdot OH + OH^- \tag{7}$$

$$Gd_2O_3(h_{vb}^+) + (H_2O) \rightarrow \cdot OH + H^- \tag{8}$$

$$O_2^- + H_2O_2 \rightarrow \cdot OH + OH^- + O_2 \tag{9}$$

$$\text{Azo dye} + \cdot OH \rightarrow \text{Products of photocatalytic degradation} \tag{10}$$

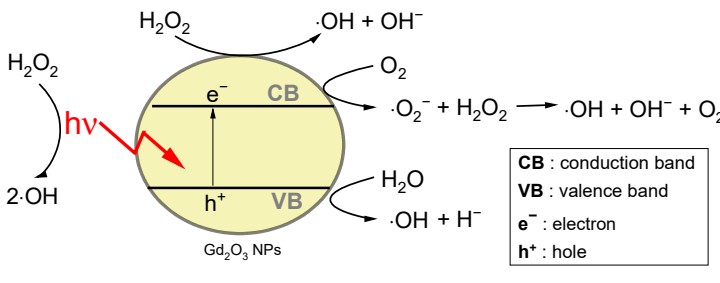

**Figure 8.** Mechanism of photocatalytic degradation of azo dyes by $Gd_2O_3$ nanoparticles.

The synergetic behavior is understandable considering the electron-capturing role of the hydrogen peroxide as an external electron hole recombined with the gadolinium oxide surface. However, it can also improve the hydroxyl radical introduction in the reaction [6,31].

### 2.3. Kinetics Study for Photocatalytic Degradation of Azo Dyes

In a first-order reaction, the rate ($\nu$) of the reaction

$$c[A] \rightarrow \text{products (A is reactant: azo dye)} \tag{11}$$

the rate is expressed as follows:

$$\nu = -\frac{dC}{dt} = kt \tag{12}$$

where $c$ is the concentration of reactant A, $k$ is the first-order rate constant, and $t$ is time. The rate depends on the concentration of the reactant A present at time, $t$. In photocatalytic degradation of azo dyes, the integrated form of rate Equation (12) may be written as in Equation (16). Separating variables of Equation (12) and taking an integral calculus,

$$\nu = -\frac{1}{C}dC = kt \tag{13}$$

$$-\ln C = kt + \text{Integration constant} \tag{14}$$

The value of the integration constant is determined when time is zero ($t$ = 0) of the following expression:

$$-\ln C_0 = \text{Integration constant (at } t \ = \ 0) \tag{15}$$

Therefore, by Equations (14) and (15), we can obtain Equation (16) [32].

$$\ln \frac{C}{C_0} = -kt \tag{16}$$

where $C_0$ is the concentration of the reactant (azo dye) at an initial time, $t = 0$, $C$ is the concentration of the reactant (azo dye) at a specific time $t$, and $k$ is the first-order rate constant [33]. The linear behavior of the curves confirms that the photocatalytic degradation of azo dyes, which were MO, AO7, and AY23, followed a pseudo first-order reaction rate law. As it can be observed from Figure 9, the values of $R^2$ (coefficient of determination) for the pseudo first-order reaction kinetics were (a) 0.99888, (b) 0.99867, (c) 0.99555.

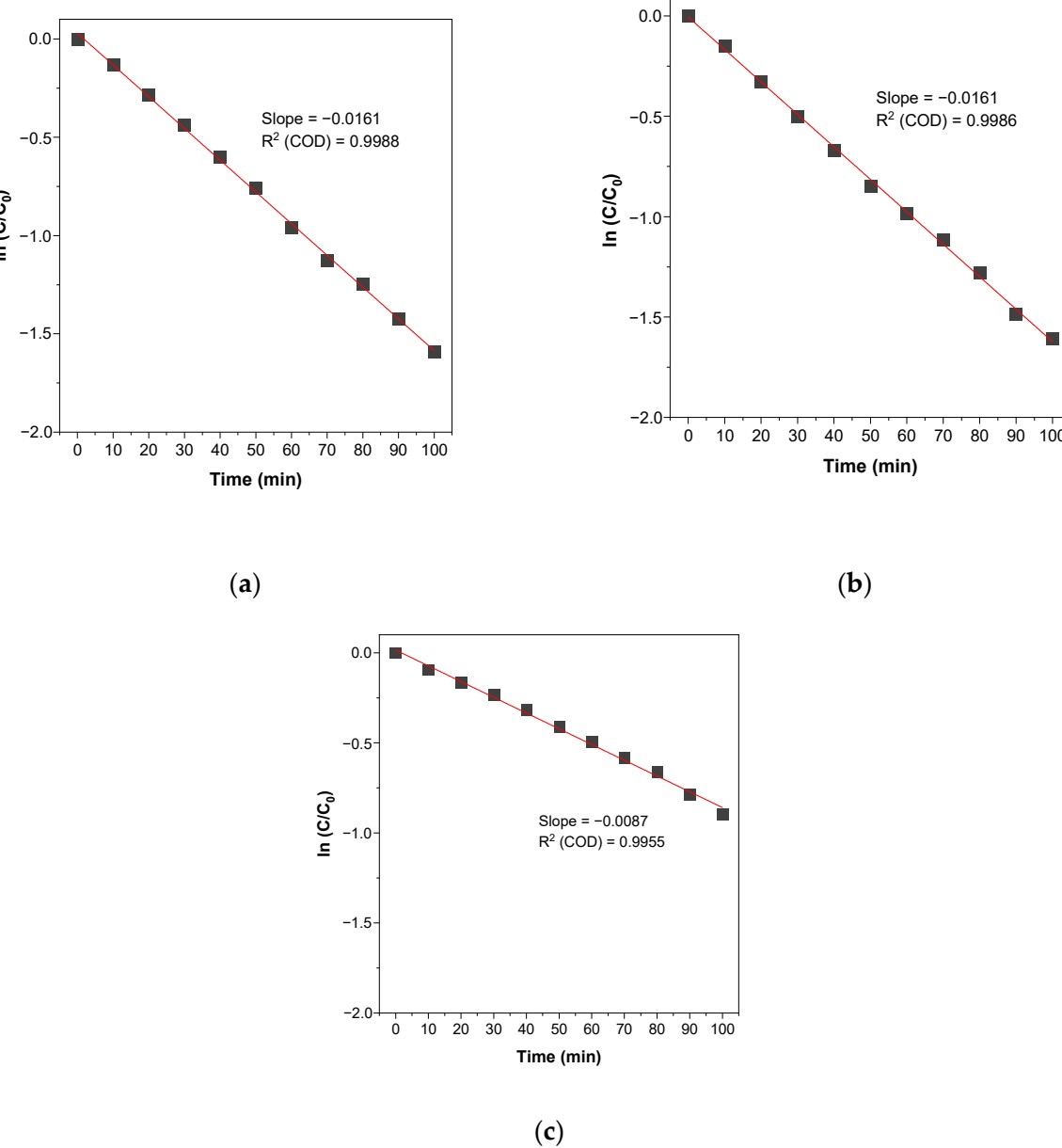

**Figure 9.** Kinetic study of the photocatalytic degradation of the azo dyes (**a**) MO, (**b**) AO7, (**c**) AY23 by the oxidation process. (Experimental conditions: initial dye concentration 0.042 mM; wavelength of UV irradiation: 254 nm; $H_2O_2$ concentration: 4.8 M; photocatalyst concentration: 1.0 g/L.).

### 2.4. Effect of Gd₂O₃ Nanoparticle Dose for Photocatalytic Degradation of Azo Dyes

The effect of the $Gd_2O_3$ dose on the photocatalytic degradation of azo dyes was investigated. An increase in the dose of the catalyst increased the photocatalytic efficiency and generated a large number of electron-hole pairs. A large number of hydroxyl radicals were available for azo dye degradation. Under the same conditions, the photocatalytic degradation of MO by $Gd_2O_3$ nanoparticles improved the reaction rate constant from $-0.00379$ to $-0.02855$ $min^{-1}$ by increasing the catalyst dose from 0 to 1.5 g/L after a 100 min degradation reaction. Under the same conditions, the photocatalytic degradation of AO7 by $Gd_2O_3$ nanoparticles improved the reaction rate constant from $-0.00060$ to $-0.02454$ $min^{-1}$ by increasing the catalyst dose from 0 to 1.5 g/L after a 100 min degradation reaction. The photocatalytic degradation of AY23 by $Gd_2O_3$ nanoparticles improved the reaction rate constant from $-0.00030$ to $-0.03712$ $min^{-1}$ by increasing the catalyst dose from 0 to 1.5 g/L after a 100 min degradation reaction. The photocatalytic degradation rate and removal% of the aqueous azo dye solution increased when the catalyst dose also increased. Under the same conditions, the degradation rate of azo dyes increased when the adsorption of the dye molecules on the catalyst surface was increased (Figure 10, Table 1).

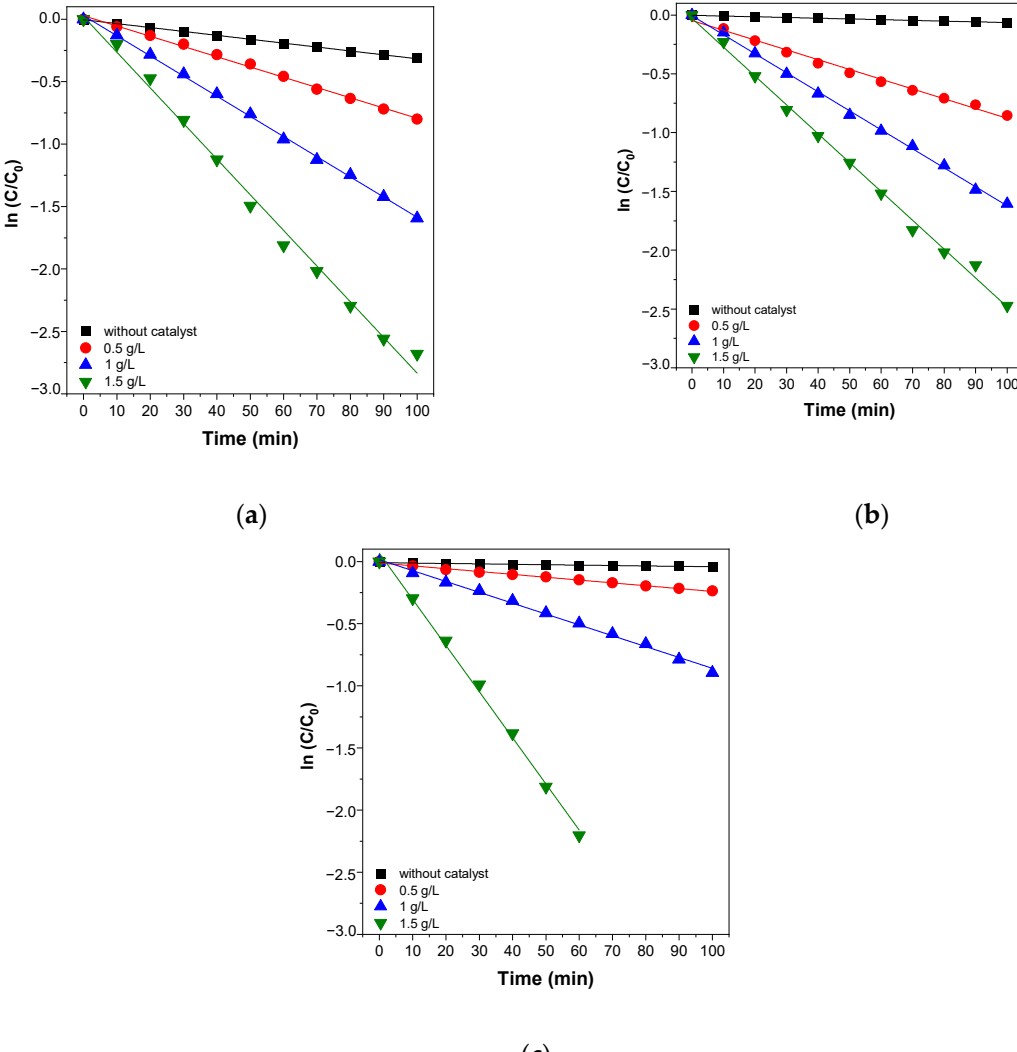

**Figure 10.** Effect of catalyst dose on the photocatalytic degradation of the azo dyes (**a**) MO, (**b**) AO7, and (**c**) AY23 by $Gd_2O_3$ nanoparticles in the presence of $H_2O_2$ at pH 5.5. (Experimental conditions: initial dye concentration 0.042 mM; wavelength of UV irradiation: 254 nm; $H_2O_2$ concentration: 4.8 M; photocatalyst concentration: 0–1.5 g/L.).

**Table 1.** Reaction rate constant ($k$) and photocatalytic degradation efficiencies (%) of $Gd_2O_3$ nanoparticles for the degradation of the azo dyes MO, AO7, AY23 in the presence of $H_2O_2$ at different catalyst doses.

| Azo Dye | Dose of Catalyst (g/L) | Reaction Rate Constant $k$ ($min^{-1}$) | Photocatalytic Degradation Efficiency (%) |
|---|---|---|---|
| Methyl Orange | 0 | −0.00379 | 23.62 |
| | 0.5 | −0.00821 | 55.03 |
| | 1 | −0.01615 | 79.68 |
| | 1.5 | −0.02855 | 93.14 |
| Acid Orange 7 | 0 | −0.00060 | 6.11 |
| | 0.5 | −0.00831 | 57.48 |
| | 1 | −0.01617 | 79.91 |
| | 1.5 | −0.02454 | 91.57 |
| Acid Yellow 23 | 0 | −0.00030 | 3.81 |
| | 0.5 | −0.00229 | 21.11 |
| | 1 | −0.00874 | 59.14 |
| | 1.5 | −0.03712 | 88.97 |

Experimental conditions: initial dye concentration 0.042 mM; wavelength of UV irradiation: 254 nm; $H_2O_2$ concentration: 4.8 M; photocatalyst concentration: 0–1.5 g/L.

### 2.5. Effect of Initial Dye Concentration on the Photocatalytic Degradation of Azo Dyes

The effect of the initial azo dye concentration on the photocatalytic degradation of azo dyes was investigated. The photocatalytic degradation experiments were carried out at different azo dye concentrations and at a constant $H_2O_2$ concentration. The initial azo dye concentration varied from 0.042 to 0.126 mM. At high initial dye concentrations, the path length of the photons entering the solution decreased and the number of ·OH radicals generated in the solution also decreased (Figure 11, Table 2).

**Table 2.** Reaction rate constant ($k$) and photocatalytic degradation efficiencies (%) of $Gd_2O_3$ nanoparticles for the degradation of the azo dyes MO, AO7, AY23 in the presence of $H_2O_2$ at different initial dye concentrations.

| Azo Dye | Initial Concentration (mM) | Reaction Rate Constant $k$ ($min^{-1}$) | Photocatalytic Degradation Efficiency (%) |
|---|---|---|---|
| Methyl Orange | 0.042 | −0.01615 | 79.68 |
| | 0.084 | −0.01214 | 69.75 |
| | 0.126 | −0.00854 | 57.28 |
| Acid Orange 7 | 0.042 | −0.01617 | 79.91 |
| | 0.084 | −0.01379 | 74.14 |
| | 0.126 | −0.01089 | 65.56 |
| Acid Yellow 23 | 0.042 | −0.00874 | 59.14 |
| | 0.084 | −0.00176 | 16.67 |
| | 0.126 | −0.00090 | 8.92 |

Experimental conditions: initial dye concentration 0.042–0.126 mM; wavelength of UV irradiation: 254 nm; $H_2O_2$ concentration: 4.8 M; photocatalyst concentration: 1.0 g/L.

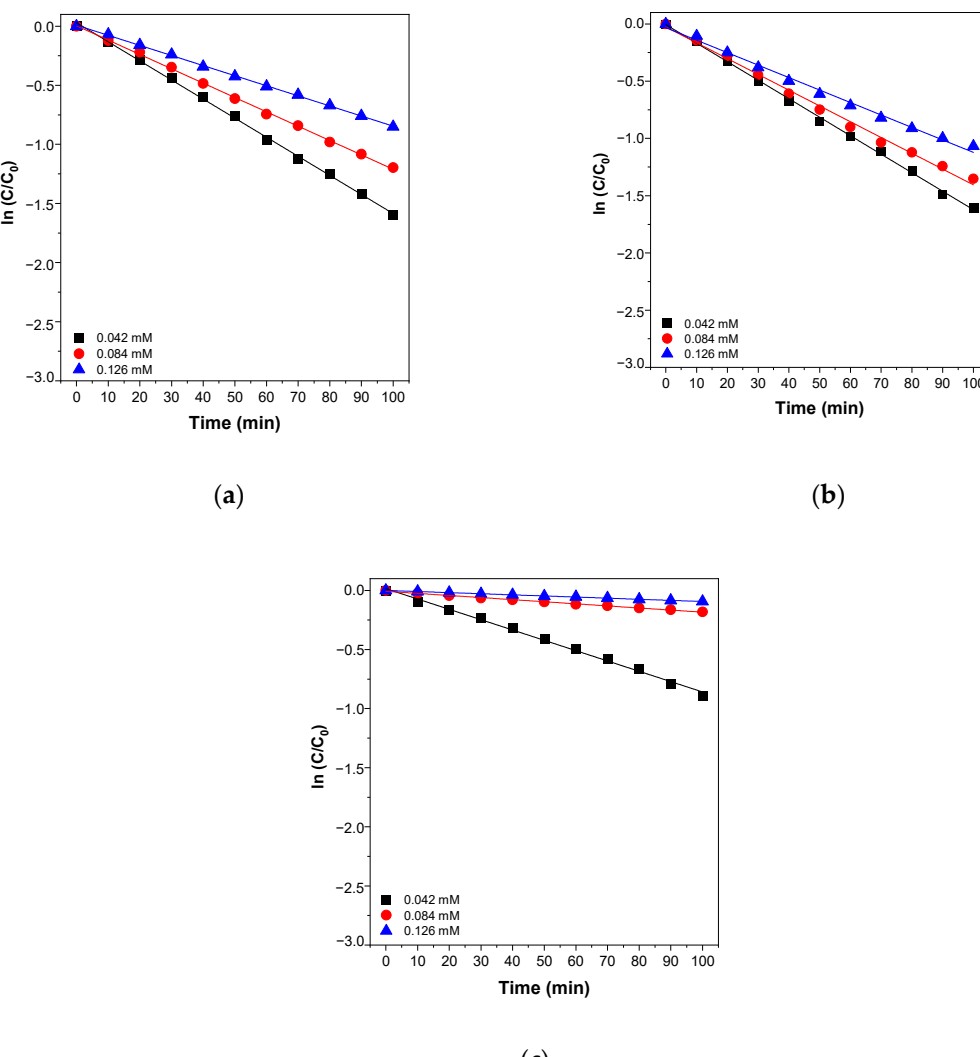

**Figure 11.** Effect of initial dye concentration on the photocatalytic degradation of the azo dyes: (**a**) MO, (**b**) AO7, and (**c**) AY23 by $Gd_2O_3$ nanoparticles in the presence of $H_2O_2$ at pH 5.5. (Experimental conditions: initial dye concentration 0.042-0.126 mM; wavelength of UV irradiation: 254 nm; $H_2O_2$ concentration: 4.8 M; photocatalyst concentration: 1.0 g/L.).

### 2.6. Effect of UV Light Wavelength on Azo Dye for Photocatalytic Degradation

In the presence of the $Gd_2O_3$ photocatalyst, UV irradiation at 365 nm (UV-A) caused higher degradation of the azo dyes than that at 254 nm (UV-C) with reduced energy consumption. For the same amount of photocatalyst, the photocatalytic degradation efficiency of MO, AO7, and AY23 obtained with UV-A radiation was higher than that obtained with UV-C [25,34,35]. This can be attributed to the higher generation rate of ·OH radicals and, consequently, the higher reaction rates at 365 nm [36]. UV-A photons penetrate much deeper into the solution and degrade a large number of dye molecules on the surface of $Gd_2O_3$ [37]. It was found that the percentage of absorbance for UV light increased significantly in the UV-A and visible regions, which favored the photocatalytic performance of the catalyst (Figure 12, Table 3) [38].

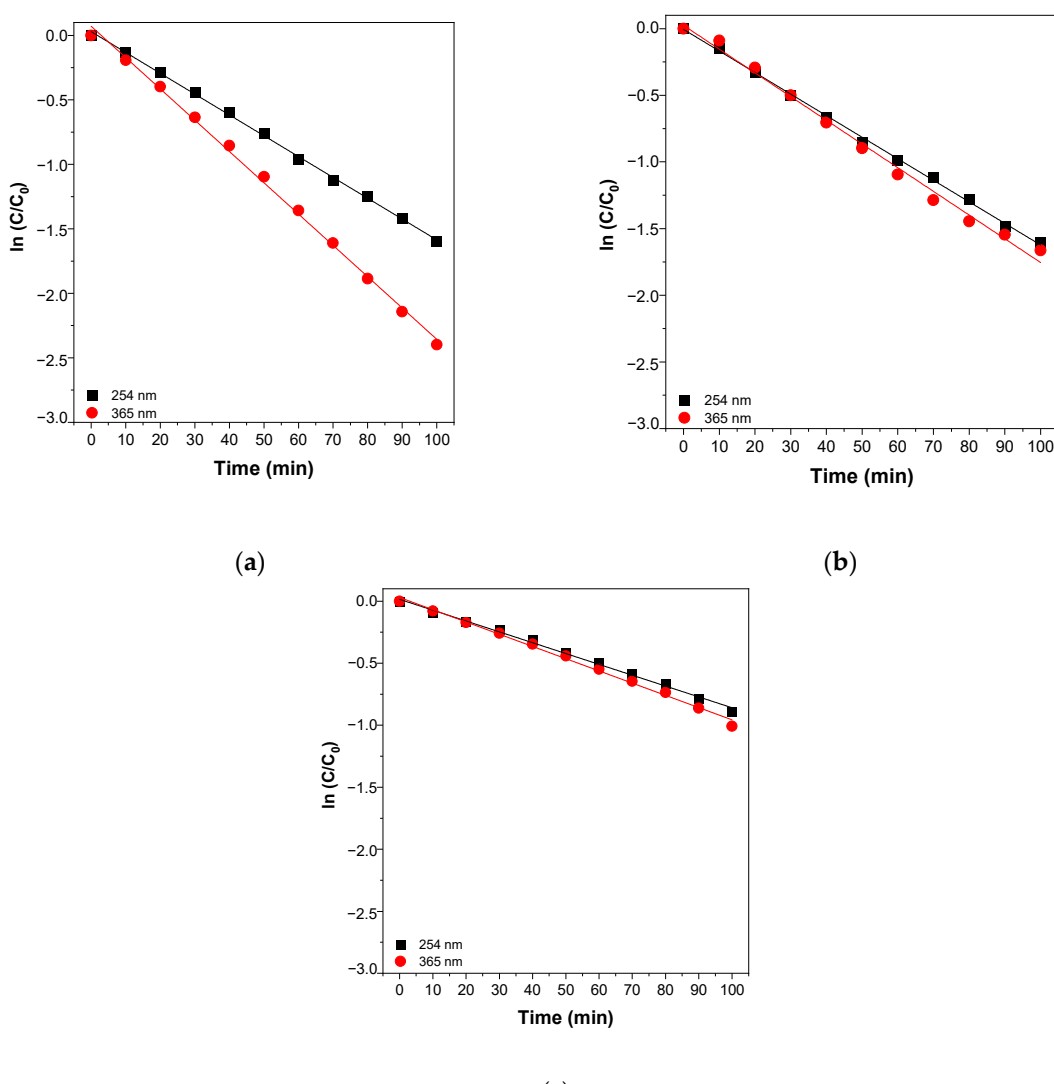

**Figure 12.** The effect of UV radiation wavelength on kinetics of the photocatalytic degradation of the azo dyes: (**a**) MO, (**b**) AO7, and (**c**) AY23 by $Gd_2O_3$ nanoparticles in the presence of $H_2O_2$ at pH 5.5. (Experimental conditions: initial dye concentration 0.042 mM; wavelength of UV irradiation: 254 nm or 365 nm; $H_2O_2$ concentration: 4.8 M; photocatalyst concentration: 1.0 g/L.).

**Table 3.** Reaction rate constant (*k*) and photocatalytic degradation efficiencies (%) of $Gd_2O_3$ nanoparticles for the degradation of the azo dyes MO, AO7, AY23 in the presence of $H_2O_2$ at different UV wavelengths.

| Azo Dye | Wavelength of UV Lamp (nm) | Reaction Rate Constant $k$ ($min^{-1}$) | Photocatalytic Degradation Efficiency (%) |
|---|---|---|---|
| Methyl Orange | 254 | −0.01615 | 79.68 |
| | 365 | −0.02427 | 90.9 |
| Acid Orange 7 | 254 | −0.01617 | 79.91 |
| | 365 | −0.02389 | 81.05 |
| Acid Yellow 23 | 254 | −0.00874 | 59.14 |
| | 365 | −0.00986 | 63.49 |

Experimental conditions: initial dye concentration 0.042 mM; wavelength of UV irradiation: 254 nm or 365 nm; $H_2O_2$ concentration: 4.8 M; photocatalyst concentration: 1.0 g/L.

### 2.7. Reusability of Gd₂O₃ Nanoperticles as Photocatalyst for Degradation of MO

The recyclability of the photocatalyst is a significant factor in its industrial application [39,40]. The recyclability of the Gd₂O₃ nanoparticles was studied by repeating the experiment five times for the photocatalytic degradation of MO solution (Figure 13). The recyclability of the Gd₂O₃ photocatalyst demonstrates a little decrease in the photodegradation of MO dye solution for each cycle, but no apparent Gd₂O₃ photocatalyst deactivation after five cycles [2,18]. This is because, after the photocatalytic degradation of MO dye, the photocatalyst was rinsed with distilled water, but a little bit of the MO dye remained on the photocatalyst's surface. As a result, the reusability of photocatalyst efficiency for the MO dye degradation was reduced, as shown in Figure 13.

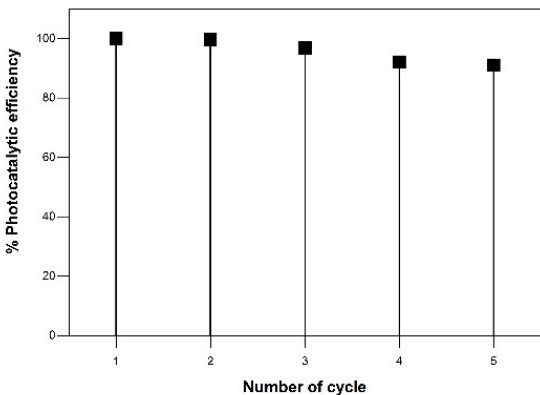

**Figure 13.** Reusability of photocatalytic degradation for MO using Gd₂O₃ nanoparticles as a catalyst.

### 3. Materials and Methods

#### 3.1. Materials

Methyl orange (MO), acid orange 7 (AO7), acid yellow 23 (AY23), gadolinium nitrate hexahydrate ($Gd(NO_3)_3 \cdot 6H_2O$), ethylamine ($C_2H_5NH_2$), and hydrogen peroxide ($H_2O_2$, 30%, *w/w*) were purchased from YAKURI PURE CHEMICAL, Tokyo, Japan), Sigma-Aldrich (St. Louis, MO, USA), Merck (KGaA, Darmstadt, Germany), and DAEJUNG CHEMICALS (Siheung, Korea). All of the chemicals were used as received, without further purification.

#### 3.2. Methods

The photocatalytic degradation of the azo dyes was carried out using a UV lamp (4 W, 254 nm; 4 W, 365 nm) and was confirmed by ultraviolet-visible (UV-Vis) spectroscopy (Shimazu UV-1601 PC, Tokyo, Japan). The morphology of the catalyst was observed by using scanning electron microscopy (SEM, JEOL Ltd., JSM-6510, Tokyo, Japan) at an acceleration voltage of 10 kV. The vibrational state of the catalyst was investigated by Raman spectroscopy at 532 nm wavelength (BWTEK i-Raman Plus, Newark, DE, USA). Fourier-transform infrared (FT-IR) spectroscopy (Thermo Scientific Nicolet iS10, Madison, WI USA) was used to obtain information on the functional groups in the catalyst. The surface elemental composition and their ionic states was analyzed using X-ray photoelectron spectroscopy (XPS) (NEXSA, Thermo Fisher Scientific USA). The crystal structures and average crystallite size of the catalysts were analyzed using powder XRD (Bruker, D8 Advance, Karlsruhe, Germany). The morphology and elemental composition of the catalyst were measured using a transmission electron microscope (TEM) (NEO ARM, JEOL, Japan) with energy dispersive X-ray spectroscopy (EDX), respectively.

#### 3.2.1. Synthesis of Gd₂O₃ Nanoparticles

The Gd₂O₃ nanoparticles were synthesized using the following hydrothermal procedure. First, 0.5 mL of $C_2H_5NH_2$ was added slowly to a beaker containing an aqueous

solution of 0.08 M $Gd(NO_3)_3 \cdot 6H_2O$ under a vigorous stirring solution. The white precipitate solution obtained was heated at 120 °C for 12 h in an oven. The resulting mixture solution was centrifuged several times and then washed with deionized water to obtain gadolinium hydroxide, which was dried at 80 °C in an oven. $Gd(OH)_3$ was annealed in an electric furnace at 700 °C for 4 h in an Ar atmosphere to obtain $Gd_2O_3$ [24].

### 3.2.2. Photocatalytic Activity of $Gd_2O_3$ Nanoparticles for Degradation of Azo Dyes Such as MO, AO7, and AY23

Stock dye solutions were prepared using 2 mM of the azo dye powders. Next, 0.5 g/L of $Gd_2O_3$ was used for the photocatalytic degradation of each 0.042–0.126 mM azo dye aqueous solution containing $H_2O_2$. The pH of the dye solutions was fixed at 5.5.

The solution was stirred with a magnetic bar for 30 min in the absence of light to attain an adsorption-desorption equilibrium between the dye molecules and the catalyst.

The photocatalytic degradation process was monitored by using UV-vis spectroscopy. Equation (3) was used to calculate the percentage of photocatalytic degradation of azo dyes. Additionally, the reaction kinetics and factors affecting the photocatalytic degradation of azo dyes were examined.

### 3.2.3. Photocatalytic Activity of Degradation of Azo Dyes Uner UV Irradiation at 254 nm and 365 nm

The photocatalytic degradation of MO, AO7, and AY23 by $Gd_2O_3$ nanoparticles was carried out in aqueous dye solutions under UV irradiation at 254 nm and 365 nm. The initial concentration of azo dye solutions was 0.042 mM and the initial $H_2O_2$ concentration was 4.8 M. The $Gd_2O_3$ nanoparticles were added to the azo dye solutions, which were irradiated with UV light (output power: 4 W–254 nm, 4 W–365 nm) for 10, 20, 30, and 100 min.

### 3.2.4. Reusability of $Gd_2O_3$ Nanoperticles as Photocatalyst for Degradation of MO

The photocatalytic degradation of MO with $Gd_2O_3$ nanoparticles was performed by repeating the experiments five times. The photocatalytic experiment was carried out under the following conditions: a 0.042 mM initial MO concentration; a 4.8 M $H_2O_2$ concentration; a 1.0 g/L photocatalyst dose; and a 100 min reaction time. After the photocatalytic degradation of MO using the $Gd_2O_3$ nanoparticles as a catalyst, the photocatalyst of the $Gd_2O_3$ nanoparticles was separated from the reaction mixture by centrifugation, washed with distilled water, and dried in an oven. Experiments on recycling were also carried out under the same conditions. To compensate for the catalyst loss during the washing process, a constant amount of catalyst concentration was maintained in each cycle test, and numerous degradation tests were conducted simultaneously at each recycle test, resulting in adequate catalysts being collected.

## 4. Conclusions

Under UV light irradiation, photocatalytic degradation of azo dyes such as MO, AO7, and AY23 was conducted in aqueous solutions with the $Gd_2O_3$ nanoparticles and $H_2O_2$. The generation of ·OH and $\cdot O_2^-$ radicals, which can oxidize the azo dye molecules, causes photocatalytic degradation of the azo dyes on the catalyst surface. The degradation rate of the azo dyes increased when the initial dye concentration decreased and when the catalyst dose increased. For the degradation of azo dyes, the photocatalytic effectiveness of the $Gd_2O_3$ nanoparticles declined in the following order: MO > AO7 > AY23. The photocatalytic degradation of the azo dye solutions under UV irradiation at 365 nm was higher than that under UV irradiation at 254 nm and was found to follow the pseudo first-order reaction rate law. The photocatalytic degradation of azo dye aqueous solutions was increased by increasing the UV irradiation time. More hydroxyl radicals are generated when $Gd_2O_3/H_2O_2$/UV light is combined, which improves the photocatalytic degradation of azo dyes. The reusability results demonstrate a small decrease in photocatalytic degra-

dation of the azo dye solution for each cycle, with no apparent decrease in photocatalytic degradation after five cycles, demonstrating the stability of the $Gd_2O_3$ nanoparticles.

**Supplementary Materials:** The following are available online at https://www.mdpi.com/article/10.3390/catal11060742/s1, Figure S1: UV-vis absorption spectra of photocatalytic degradation of the MO dye (a) $UV/H_2O_2$ (b) $H_2O_2/Gd_2O_3$ (c) $UV/H_2O_2/Gd_2O_3$ (d) $UV/Gd_2O_3$ methods within the time period of 100 min. (Experimental conditions: initial dye concentration 0.042 mM; wavelength of UV irradiation: 254 nm; $H_2O_2$ concentration: 500 mM; photocatalyst concentration: 1.0 g/L.), Figure S2: UV-vis absorption spectra of photocatalytic degradation of the AO7 dye (a) $UV/H_2O_2$ (b) $H_2O_2/Gd_2O_3$ (c) $UV/H_2O_2/Gd_2O_3$ (d) $UV/Gd_2O_3$ methods within the time period of 100 min. (Experimental conditions: initial dye concentration 0.042 mM; wavelength of UV irradiation: 254 nm; $H_2O_2$ concentration: 500 mM; photocatalyst concentration: 1.0 g/L.), Figure S3: UV-vis absorption spectra of photocatalytic degradation of the AY23 dye (a) $UV/H_2O_2$ (b) $H_2O_2/Gd_2O_3$ (**c**) $UV/H_2O_2/Gd_2O_3$ (d) $UV/Gd_2O_3$ methods within the time period of 100 min. (Experimental conditions: initial dye concentration 0.042 mM; wavelength of UV irradiation: 254 nm; $H_2O_2$ concentration: 500 mM; photocatalyst concentration: 1.0 g/L.), Figure S4: UV-vis absorption spectra of the (a) MO, (b) AO7, (c) AY23 under 254 nm irradiation and (d) MO, (e) AO7, (f) AY23 under 365 nm irradiation within the time period of 100 min. (Experimental conditions: initial dye concentration 0.042 mM.) Figure S5: UV-vis absorption spectra of photocatalytic degradation of the (a) MO, (b) AO7, (c) AY23 by $UV/H_2O_2$ methods under 254 nm irradiation and (d) MO, (e) AO7, (f) AY23 under 365 nm irradiation within the time period of 100 min. (Experimental conditions: initial dye concentration 0.042 mM; $H_2O_2$ concentration: 4.8 M)

**Author Contributions:** Conceptualization, W.-B.K.; methodology, S.J., J.-W.K.; writing—original draft preparation, S.J., J.-W.K.; writing—review and editing, W.-B.K. All authors have read and agreed to the published version of the manuscript.

**Funding:** This research was supported by research foundation of Sahmyook University.

**Data Availability Statement:** In the data presented in this study are available in article.

**Conflicts of Interest:** The author declare that they have no conflict of interest.

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
