# Peer review of "Synthesis of Gd2O3 Nanoparticles and Their Photocatalytic Activity for Degradation of Azo Dyes"

_catalysts, doi:10.3390/catal11060742_

Round 1
Reviewer 1 Report
The manuscript deals with the photocatalytic activity of Gd2O3 nanoparticles combined with H2O2 in the degradation of three azo dyes. First, authors synthetize and characterize the nanoparticles and later they study their ability to degrade the azo dyes, analyzing several variables (Gd2O3 dose, initial dye concentration and UV light wavelength). The manuscript has been considerably improved in comparison with the first version, and from my point of view, it could be considered for publication. However, some minor revision would improve the manuscript:
Line 58: MO, AO7, and AY23 are named by the abbreviature, however, before the abbreviatures, I could not find in the main text the name of the compounds. They appear in the abstract, but not in the main the text. Since they appear in this line for the first time, I would suggest including their names here for the sake of clarity. Actually, this detail is in line 270, but please include it at the beginning of the manuscript.
Line 109: Authors say: “…the azo dye degradation process was very slow”. I guess that it is based on Figure S4, the control experiment. Please cite the corresponding figure of the SI in the main text. Regarding to this Figure S4, I think that there is a mistake in the footnote, since it says the “photocatalytic degradation” but if I understood well, this is a control and no photocatalyst was employed, therefore “photocatalytic degradation” should be removed.
Line 111: Authors say: “…containing H2O2 showed” and they refer to figure 6 where H2O2 and Gd2O3 are used in combination. I would suggest to say “…containing H2O2 and Gd2O3 showed” for the sake of clarity. Since there are several control experiments with different combinations, it can be confusing.
Line 117: I guess that the 26.62% of degradation is referred to figure S1 a. Please, cite the figure in the main text in order to understand the figures of the SI. Furthermore, I would suggest to explain the other control experiments performed for MO (Figure S1) at this point, since the % of degradation is indicated later in the histograms of Figure 7 but one cannot know the origin of the data.
Lines 118 – 120: As in the case of MO, I guess that the control experiments are demonstrated with figure S2, but please, link the origin of the percentages, i.e. 6.11% (line 119), with the figure of the SI.
Lines 122-124: As in the other cases, please, link the origin of the percentages with the figure of the SI.
Figure 8: In the pdf version that I received the “points (·)” to indicate radicals appear as unknown symbols. Please check it.
Lines 197 and 200: Please check the rate constants units.
Line 238: As in the previous version, what seem experimental results are followed by literature references. From my point of view, this is confusing since the information can be attributed to both, the experimental results and the references. I would suggest rephrasing the sentence in order to differentiate what is in the literature and what has been contributed by the authors.
Line 306-307: Authors monitor the reaction though UV-vis spectroscopy, and from that data, they calculate the % of degradation. I guess that they properly consider the adsorbed azo dye before the calculus of the degradation%, however they do not describe how they calculate the % of degradation. I think it is not so obvious and should be described for the sake of clarity.
Lines 325 – 328: It is not clear to me how do they compensate the catalyst loss. In the first version authors say that they added fresh catalyst (but they did not specify the rate of new/old). In this new version, as far as I understand, they perform a first degradation reaction in multiple flasks (in parallel) and collect the catalyst after the reactions. With all the photocatalyst collected, they can wash it and have enough amount to perform the next cycle. I would suggest rephrasing the sentence in a clearer way for an easier understanding. Besides, I have some concerns with this methodology: i) If the photocatalyst is washed only with water, how can it desorb the azo dyes? Washing with water provides a medium equivalent to the degradation reaction (aqueous media), thus, if in the reaction medium the catalyst adsorbs the azo dyes, why does it desorb them in the washing process? ii) A possible explanation to this could be that the washing process does not desorbs the azo dyes, in this case, I would expect that the catalyst surface saturates due to the exposure to more and more azo dyes, but its activity seems to be very good according to Figure 13. I think that all these matters should be properly discussed by the authors for the sake of clarity in the reusability of Gd2O3 section (2.9, line 245).
Line 422: I could not find reference 30 in the main text. Please locate the reference in the main text or remove it from the list.
Finally, I do not usually say this, but there are several mistakes related to language, for example, in the footnotes of figure 6, 7, 10,… it says “wavelength of irradiated UV”, I guess it is “wavelength of UV irradiation”. I would suggest revising the manuscript in terms of language.
Author Response
Dear Reviewer 1
Thank you for your comments on my manuscript.
Please see the attachment.

Reviewer 2 Report
This is indeed an interesting study, which has synthesized Gadolinium oxide (Gd2O3) nanoparticles were prepared via the reaction of gadolinium nitrate hexahydrate (Gd (NO3)3•6H2O) and ethylamine (C2H5NH2). The chemical and physical properties of the system were examined by using scanning electron microscopy(SEM), trans-mission electron microscopy(TEM), energy dispersive X-ray(EDX) spectroscopy, X-ray diffraction (XRD), X-ray photoelectron spectroscopy(XPS), Raman spectroscopy, Fourier-transform infrared (FT-IR) spectroscopy, and Ultraviolet-Visible (UV-vis) spectroscopy. It was observed that at a constant H2O2 concentration, the photocatalytic degradation efficiency of the Gd2O3 nanoparticles for various azo dyes was in the order: methyl orange > acid orange 7 > acid yellow 23, and that the degradation efficiency azo dyes was followed a pseudo first-order reaction rate law.
Whereas the chemical reactions, together with pictures and discussions provided may be good, it is not a well prepared ms. Before I recommend publication of this study, I ask the authors to fix the following issues.
1- Fonts sizes on page 2 are carelessly given.
2- A transparent objective of the paper is not carefully written in the introduction section of the ms. In fact, it is largely missing.
3- How come a bond can show degradation, as written "The azo bond (-N=N-) showed a degradation of ..."?
4- Can first principles results of band structure be presented in a revision?
5- Background references and conclusion section are not appropriate. Revision is necessary for further improvement of both references and quality of writing.
6- Writing quality of the entire ms should be improved as scientific usages and grammar are not appropriate.
Author Response
Dear Reviewer 2
Thank you for your comments on my manuscript.
Please see the attachment.

This manuscript is a resubmission of an earlier submission. The following is a list of the peer review reports and author responses from that submission.
Round 1
Reviewer 1 Report
The manuscript deals with the photocatalytic activity of Gd2O3 nanoparticles combined with H2O2 in the degradation of three azo dyes. First, authors synthetize and characterize the nanoparticles and later they study their ability to degrade the azo dyes. Although the idea is adequate and the use of Gd2O3 nanoparticles could be interesting, some results do not agree with previous studies reported in the literature and there is no discussion to explain these differences. Besides, the presentation of the results/methods could be improved in some sections, some extra control experiments would shed some light to the reported data, some references do not seem to match with the text and there are several spelling mistakes. Unfortunately, within all this, I cannot recommend the manuscript for a Journal such as Catalyst with an IF of 3.520.
However, I think that the manuscript could be improved following the next suggestions:
In general terms:
Regarding to the results provided in the literature, it has been stablished that combination of UV light with H2O2, promotes the generation of hydroxyl radicals (OH·), which are able to degrade (very effectively) organic compounds, such as the azo dyes. For example, see C. Galindo et al. Journal of Photochemistry and Photobiology A: Chemistry 130 (2000) 35–47 (385 cites) or A. Aleboyeh et al. / Dyes and Pigments 57 (2003) 67–75 (119 cites). This background is completely different from what authors propose in the first columns of the three histograms of figure 7, where the degradation efficiency of UV light + H2O2 is quite low. Since the use of UV light + H2O2 is a well stablished procedure in AOPs, these differences should be properly discussed. In this sense, mechanism proposal (page 7) do not include the formation of the hydroxyl radical from UV light + H2O2 (and other possible reaction), see specific comment of line 178. Besides, the UV-vis spectra employed to do the histograms of figure 7 should be included as part of the supporting information. Finally, since the azo dyes are able to absorb at the irradiation wavelengths (254 nm and 365 nm), control experiments should be included to monitor direct photolysis.
Regarding to the presentation of the results there are several issues: i) The section Materials and Methods is not properly divided, and several procedures should be included at this section (see specific comments for lines 109-128 and line 276). ii) In many experiments descriptions, references create some confusion between the information provided by the reference and the extracted from the experiment (see specific comments for lines 135-148 or 159-161 although there may be some more). iii) The footnotes for the figures and tables should provide more detailed information, as indicated below.
In the next lines, some specific suggestions are included:
Lines 27 - 29: Authors mention “the advanced oxidation process (AOP)” as a unique process. This sentence may be confusing since nowadays there are several AOPs: “One of the first references to AOPs was by Glaze in 1987 as processes that “involve the generation of hydroxyl radicals in sufficient quantity to affect water purification”. The definition and development of AOPs have evolved since the 1990s and include a variety of methods for generating hydroxyl radical and other reactive oxygen species including superoxide anion radical, hydrogen peroxide, and singlet oxygen. However, hydroxyl radical is still the species most commonly tied to the effectiveness of AOPs.” J. Phys. Chem. Lett. 2012, 3, 15, 2112–2113. I recommend reformulating the sentences.
Line 29: Since there are several AOPs I would suggest specifying the methodology that authors are referring with reference 5 (I guess it is the use of TiO2) or indicate that there are more oxidizing species than just hydroxyl radical and superoxide radical anion. In addition, in this line, I think there is a mistake: instead of dioxide radicals I guess it is superoxide radical anion (O2·-)
Line 43: I guess that Ref.12 deals with the comparison of Gd2O3 vs Gd2O3 dopped with Pd, thus Gd2O3 nanospheres are not used as an additional oxidizing agent, as the main text suggest. Besides, Ref.13 does not involve the use of Gd2O3. Please, check the references.
Line 51: Authors say “The addition of hydrogen peroxide increased the rate constants of heterogeneous oxidation process.” It is not clear to me if it is a general statement (then it should be written as “the addition of hydrogen peroxide increasES the rate”….) or if it is something from a specific study of the review of Ref.4 (then it should explain a little bit the idea of the reference). Please, rephrase the sentence to be more specific.
Line 76: I guess authors mean “compounds” instead of “compositions of”.
Line 105 - 106: Authors say “extrapolating the linear portion of the curve in the UV-vis absorbance spectrum (Figure 5(a))”, I guess that the extrapolation is not in the UV-vis absorbance spectrum, figure 5a, but in figure 5b.
Line 109 – 128: All the descriptions included at these lines would seem more likely to be as part of the “Materials and Methods” sections instead of part of the “Results” section, since they describe the experiments but not the results. Please move the explanations to the “Materials and Methods” section for the sake of clarity.
Line 132: Authors mention the performance of a control reaction in the absence of catalyst. Since they use the Gd2O3 nanoparticles with H2O2, it seems important to specify if that control was in the absence of both (Gd2O3 nanoparticles and H2O2) or just in the absence of the Gd2O3 nanoparticles. Besides, I would suggest including this control in the SI.
Lines 135 – 148: Several concerns here:
- Line 135 refers to aqueous solutions containing H2O2 and cites Figure 6, but the footnote of Figure 6 just refers to the presence of Gd2O3 I guess that both compounds (Gd2O3 nanoparticles and H2O2) are in the mixture but please specify it in both, the main text and the footnote of figure 6.
- After the explanation of each experiment, there are several references (23, 26, 27). This is a little bit confusing since it is not clear if the experiments and/or the explanations that they include have been made by the authors or if they are part of the literature. I guess that authors performed the experiments, so I would suggest rephrasing some sentences or move the references to the specific information extracted from the literature.
- Figure 6: This figure appears before of being cited at the main text. Please, move it after. Besides, I would suggest including the initial concentrations of each compounds at the footnote for the sake of clarity.
Line 149: To the best of my knowledge, Ref. 22 does not deal neither with Gd2O3 particles nor with H2O2 adsorption, the reference studies the degradation of an azo dye with TiO2, thus they study the adsorption of the azo dye onto TiO2 and besides the influence of adding H2O2.
Line 150: I guess that “capacity of metals” is “capacity on metals”.
Line 153: I guess that “photocatalytic degradation increased“ is “photocatalytic degradation increases”.
Lines 159 and 161: Again, after the explanation of each experiment there are several references (28, 29, 13, 15, 30-33). As in the other case, this is a little bit confusing since it is not clear if the experiments and/or the explanations that they include have been made by the authors or if they are part of the literature. I guess that authors performed the experiments, so I would suggest rephrasing sentences or move the references to the specific information extracted from the literature.
Figure 7: Some of the percentages indicated in the figure are those indicated in lines 141 (79.68%) and 148 (3.81%) but others do not coincide with those specified in the lines: line 141 (23.6% vs 26.62%) and line 144 (79.91% vs 79.92%). Please check the numbers. Besides, the footnote says “under different conditions”, I would suggest to include the conditions in the footnote for the sake of clarity.
Line 178: As previously mentioned, some reactions are missing here:
- Once superoxide radical anion has been formed, it can also oxidize the azo compounds.
- H2O2 + UV light à2OH. The formation of OH· through this pathway should be discussed properly since there are many references in the bibliography that use this technique to form hydroxyl radical to degrade pollutants, dyes, or organic matter in general
Line 181: I guess that “as following” is “as follows”.
Figure 9: Please, specify conditions.
Lines 203, 204 and 208: I do not understand the units of the kinetic constants (cm-1). I guess it should be min-1.
Figure 10, Table 1, Figure 12 and Table 3: Please specify H2O2 concentration.
Table 2: Please specify H2O2 and Gd2O3 nanoparticles concentration.
Lines 233 – 242: Authors say that photodegradation effectiveness is UV-A > UV-C, favoring the performance of the catalyst (line 242). However, they also admit that the greater degree of degradation can be attributed to the higher absorption of the reaction solution at 365 nm (with the same amount of photocatalyst). If the wavelength of irradiation has changed, direct photolysis of the azo dyes (direct degradation without the need of the catalyst) should not be discarded. To do such affirmation, control experiments should be performed. Therefore I would suggest to include a direct irradiation experiment of the azo dyes (with and without H2O2) in order to check the real involvement of the photocatalyst.
Line 245 -248: Authors explain that they add fresh catalyst to compensate the loss of the catalyst during the washing step, but they do not specify the ratio of fresh/reused photocatalyst. Please, specify the proportions.
Line 276: This line and the next ones (until the end of the paragraph) describe methodologies that are not part of the synthesis of the Gd2O3 nanoparticles. I would suggest a new section for these explanations for the sake of clarity.
Line 288: I guess that the dot at the end of the line is a mistake.
Line 289: I guess that “can be oxidized” is “can oxide”
Line 293 - 294: Authors say “degradation … under UV irradiation at 365 nm was higher than that under UV irradiation at 254 nm” but, as it has been previously said, no control experiments were performed, thus I would recommend to perform the direct photolysis experiments to check the real involvement of the photocatalyst.
Reviewer 2 Report
In this work, Jeon et al. prepared Gd2O3 structures for photocatalytic degradation of Azo dyes. It is an interesting work, but not very well written. Improvements must be made before it can be accepted for publication. Introduction of this paper is too short, in particular missing recent advances in Gadolinium oxide materials for various applications including photocatalytic uses.
The authors seem to have done many experiments of materials characterization and application. However, some major corrections should be made.
1. Looking at Figure 2, SEM images clearly shows that the structure is one-dimensional (1D), but the authors said Gd2O3 nanoparticles. I don't think that this structure is a particle. In the elemental mapping shown in Figure 2c, the signal for Gd is very poor. The authors should repeat this test.
2. It may look better if the authors put Raman and FTIR results in one Figure as "Figure 3a and 3b".
3. XPS should be carried on the Gd2O3 structure with high-resolution XPS of Gd.
Reviewer 3 Report
The authors present a study focused on photocatalytic degradation of different type of azo dyes over gadolinium oxide nanoparticles (synthesized by hydrothermal method) with the presence of H2O2 in solution. The catalyst has been characterized with different methods such as SEM, XRD, Raman spectroscopy, UV-vis spectroscopy, and FTIR. It has been showed that photocatalytic efficiency of Gd2O3 nanoparticles for the degradation of the azo dyes decreased in the order: MO > AO7 > AY23. And it is showed that photocatalytic performance dependence on dye concentration and catalyst dose.
My comments and questions are listed below:
Questions/Statement
In general, the manuscript has serious formatting (i.e., see line number 39) and spelling issues. The letter and number size in the graphs can be increased for better visibility. Some graphs are not visible (i.e., see figure 7 a, b, c).
Introduction
- The background and aim of the study should be indicated clearly. For example, what are the other studies on photocatalytic activity of G2O3 in aqueous environment and what is the gap has been fulfilled in study? Briefly, what is the aim and novelty of this study?
Results
- The authors showed some data on the material characteristics of Gd(OH)3, as well. However, I have not seen any photocatalytic activity result of Gd(OH)3. I kindly want to ask the authors why the material properties of Gd(OH)3 has been illustrated in the result part?
- If the optical bandgap of bulk gadolinium oxide is 5.30 eV. What is the reason of such low bandgap value (5.04 eV) of Gd2O3 particles in the current study?
Conclusion
4 . The conclusion of the study is weak. It should be improved.
The authors present a study focused on photocatalytic degradation of different type of azo dyes over gadolinium oxide nanoparticles (synthesized by hydrothermal method) with the presence of H2O2 in solution. The catalyst has been characterized with different methods such as SEM, XRD, Raman spectroscopy, UV-vis spectroscopy, and FTIR. It has been showed that photocatalytic efficiency of Gd2O3 nanoparticles for the degradation of the azo dyes decreased in the order: MO > AO7 > AY23. And it is showed that photocatalytic performance depends on dye concentration and catalyst dose.
